# Effect of Microwave Drying on the Drying Characteristics, Color, Microstructure, and Thermal Properties of Trabzon Persimmon

**DOI:** 10.3390/foods8020084

**Published:** 2019-02-23

**Authors:** Soner Çelen

**Affiliations:** Department of Mechanical Engineering, Namık Kemal University, Tekirdağ 59860, Turkey; scelen@nku.edu.tr; Tel.: +90-546-682-20-21

**Keywords:** activation energy, moisture diffusivity, microwave dryer, persimmon, scanning electron microscopy

## Abstract

In this study, changes in the drying kinetics, color change, and the energy consumption for microwave energy were investigated for Trabzon persimmon. In addition to that, the microstructure of the persimmon was also investigated by considering its thermal changes. It is important to be aware of the purpose of the drying process for determining the drying system. Results of this research showed that 460 W for 7 mm slice thickness depending on energy consumption, 600 W for 5 mm slice thickness depending on drying time, and 600 W depending on color changes were found as suitable drying processes depending on drying conditions. The effective diffusion values varied between 2.97 × 10^−8^ m^2^ s^−1^ and 4.63 × 10^−6^ m^2^ s^−1^. The activation energy values for 5 mm, 7 mm and 9 mm slice thickness were estimated as 32.82, 18.64, and 12.80 W g^−1^, respectively. The drying time and energy consumption decreased, whereas drying rate increased with an increase in the microwave energy. The number of pores increased compared to structure of fresh sample, and the pores got to be larger for 5 mm slice thickness as the power level increased. Results showed that the applied microwave energy had an important effect on the heating of the material and the change in the microstructure.

## 1. Introduction

While persimmon (*Diospyros kaki* L.) is known as “Trabzon persimmon” in Turkey, it is also known as “Asian” or “Japanese apple” in South America [1]. The persimmon has its origin in China and is grown in abundance in the Mediterranean region of Turkey [2]. Since it is a seasonal fruit, it should be consumed in a short period of time. The drying of the persimmon makes it is possible to be consumed even out of its season. Moreover, it strengthens the immune system and helps with cancer prevention [3,4]. As it is used in making jam, marmalade, ice cream, and cake, persimmon is an important resource in the food industry with its high shelf life when dried. Dried persimmons are consumed as muesli, aperitif, and breakfast cereals [5]. Due to its nutritious nature, aroma, taste, and medical value, persimmons are in high demand in international markets [6].

Drying is defined as the removal of the water or any sort of liquid in a substance. The purpose of the drying is to stop the development of microorganisms and biochemical reactions that might occur in the products by removing free water, thus enabling the product to last longer with the microorganisms which are reduced to a level at which they can reproduce no longer [5,7,8,9]. In addition, color is one of the most important sensory features that is used in determining the consumer’s acceptance of a food product [10,11].

Because of the fact that drying is used in many industries around the world, there are many different types of dryers [12]. In recent years, because of the long drying time, drying by microwave energy has been studied due to the fact that it has advantages such as low drying time, uniform heat dissipation, final product quality, and low energy consumption [13,14,15]. Drying with microwave energy differs significantly from conventional drying methods. In conventional drying, there is a gradual transmission of the heat from the surface of the material to the interior due to the temperature difference between the hot surface and the cooler interior. However, in microwave drying, the electromagnetic field affects the material as a whole, which causes the water molecules in the material to vibrate millions of times every second. This vibration and the resulting energy allow the moisture in the material to evaporate quite quickly [16,17]. Drying by microwave energy is an alternate drying method that provides advantages such as high heat conduction to the interior of the dried material, cleaning, energy recovery, energy process control quick start, and termination of the drying process [18].

The mechanism of the moisture movement in materials is shown by effective moisture diffusion. In the drying process, the diffusion is assumed to be as water diffusion on the surface of the material. In fluid mechanics and mass transfer processes, knowing the diffusion coefficient provides convenience in their design. The drying process of most of the food products takes place in the second phase of the drying, which is the reducing rate phase and moisture transfer controlled by internal diffusion [19]. Throughout the drying process, water is carried towards the surface by means of the molecular diffusion mechanism. Due to the complex process of drying, diffusion of the water in the food product for all the constant, and decrease of flux segments are defined by effective diffusion coefficient [20]. The second law of Fick is used to define moisture diffusion process [21]. The diffusion coefficient of a product is the property of that product and its value depends on the internal structure of the material. Moisture diffusion defines all possible mechanisms of moisture movement in the product such as fluid diffusion, vapor diffusion, surface diffusion, capillary flow, and hydrodynamic flow [22]. When taking a look at the literature on the published studies regarding the determination of the diffusion coefficient in different products and microwave drying; Darvishi [23] clover, Abolhasani and Ansarifar [24] white mulberry, Aghbashlo et al. [25] carrot, Arslan and Ozcan [26] onion, Celen et al. [27] pumpkin, Yogurtcu [28] apple, Ozbek, and Dadali [29] mint leaf drying can be seen. In the drying of persimmon, varying sort of dryers was used. These studies consist of; heated air drying [2], freeze drying [30], osmotic drying [10], microwave-vacuum drying [31], tray dryer [3], convective dryer [4] and solar drying [1,6]. Moreover, there are some studies on the effects of persimmon on human health, as well as its shell and leaves [32,33,34].

Persimmon is a seasonal fruit that is consumed as fresh or dried. The reason why dried persimmon is used is that it is preferred widely. But there are few studies regarding microwave energy. The aim of this study is to examine and discuss the effects of microwave energy on persimmon slices. These effects are stated as; (1) determination of the optimum power of microwave drying, (2) determination of the effective diffusivity and activation energy, (3) determination of the drying rate, (4) determination of the color changes and shrinkage in terms of the product quality, (5) calculation of the energy consumption, (6) determination of the heat change in the product and (7) micro-structural changes.

## 2. Material and Methods

Approximately the same size and type of persimmons (*Diospyros kaki* L.) were bought from a local market and stored in a freezer with a temperature of 4 °C for 1 day.

### 2.1. Preparation of the Persimmons

Prior to the start of the experiment, the persimmons were washed, peeled, and their seeds were removed. Then the persimmons were cut in slices by a knife to 5 mm (4.8–7.5 cm diameters), 7 mm (4.2–5.7 cm diameters), and 9 mm (5–7.2 cm diameters) of thickness.

### 2.2. Drying Procedure

The drying procedure was performed with the microwave dryer seen in Figure 1 (Arçelik MD 574 S, Arçelik A.Ş., Istanbul, Turkey) with the cavity dimensions of 419 × 428 × 245 mm. After these preparations (washing, seed removal, and slicing), the persimmon was placed in a glass cup with a 15 cm diameter, 1 cm depth, and the first weights were determined for each of the slices. In order to determine their weights in the dry state, they were stored in a dying oven (MINGDA KH-35A, MINGDA Technology Co., Ltd., Shenzhen, China) for 24 h at 105 ± 1 °C and its initial moisture value was determined as 3.51 (g water g dry matter^−1^) according to the dry base (d.b). Experiments were conducted at 120 W (with three min intervals), 350 W (with 1-min intervals), 460 W (with 1-min intervals) and 600 W (with 30 s intervals) and 2450 MHz frequency. Initial values for the dimensions, color properties, and thermal properties of the products were determined. A thermal camera (FLIR Ex E6, FLIR Systems, Inc., Tallinn, Estonia) was used in determining temperature changes. The drying process was ended when the samples reached the value of 0.10 (d.b). The weights of the persimmons were measured by using the below balance weighing feature of the scale Precisa XB 620M model (Precisa Instruments AG, Dietikon, Switzerland) with a precision of 0.001 g. The product was hanged stationary during the drying procedure. Drying tests were repeated 3 times for each experimental condition in order to minimize uncertainties in the results. After the drying process was over, the energy consumption values were measured with a 0.1 kWh precision energy meter (Trotec BX09, Trotec End.Ürn.Tic.Ltd.Şti, Istanbul, Turkey) and the color change values were measured with a Spec HP-200 brand color meter (SinoDevices Group Ltd., Jiangsu, China).

### 2.3. Data Analysis

#### 2.3.1. Moisture Content and Moisture Rate

Moisture content of agricultural products was determined in dry basis, which was calculated as the rate of water amount in the product to its dry weight. Moisture content (g water/g dry matter) values were calculated using Equation (1a). Dimensionless moisture ratio of persimmons during drying was generally calculated by the following Equation (1b). Since the equilibrium moisture content turned out to be too low of a value when compared to *m_t_* and *m_o_*, the value was taken as zero as given in Equation (1c) [35].
(1a)$$m_{t}=\frac{M_{w}}{M_{w}-M_{s}}$$
(1b)$$\text{MR}=\frac{m_{t}-m_{e}}{m_{o}-m_{e}}$$
(1c)$$\text{MR}=\frac{m_{t}}{m_{o}}$$

In these equations: *M_s_*: dry mass of persimmon (g), *m_o_*: initial moisture content, (g water/g dry matter), *M_w_:* water mass of persimmon (g), MR: dimensionless moisture content, m_t_: moisture content of persimmon at certain time, (g water/g dry matter), *m_e_:* equilibrium moisture content.

#### 2.3.2. Drying Rate

The change in the moisture content of the dried product in unit time is defined as “drying rate” (DR). Drying rate is calculated as in Equation (2) [22].
(2)$$\text{DR}=\frac{m_{t+\Delta t}-\;m_{t}}{\Delta t}$$
where DR is the drying rate (g water/g dry matter.min.); *m*_*t*+Δ*t*_ is the moisture content at *t*+Δ*t* (g water/g dry matter) and *t* is drying time (min).

#### 2.3.3. Shrinkage Test

Shrinkage of the food products during drying is a natural physical occurrence. Shrinkage was determined using difference between volume values before and after the drying process. Volume values were calculated using measured diameter and thickness of the slice. The percentage of volumetric shrinkage value was calculated as in Equation (3) [36].
(3)$$\mathrm{shrinkage}\;\%\;=\frac{V_{o}-V}{V_{o}}\times 100$$
where *V_o_* and *V* are the volumes of the persimmon at the beginning and at the end of the drying experiment, respectively.

#### 2.3.4. Moisture Diffusivity and Activation Energy

According to the second law of Fick, if the material properties are fixed, the shrinkage and pressure changes were not taken into account during the drying process. It is stated that the heat transfer on the product occurs by transmission inside and by natural convection on the surface of the product. The mass transfer occurring in the product was considered to be molecular diffusion on the inside, and mass transfer mechanism by forced convection on the outside. Evaporation was assumed to occur only on surfaces. The initial moisture distribution was taken as homogeneous and the moisture distribution during the process is considered symmetrical [20]. Effective moisture diffusivity was considered constant during the drying process. Flick’s diffusion equation (Equation (4)) was used to calculate effective diffusivity using the slope method for slab geometries. The second Law of Fick in one-dimensional mass transfer during different materials drying is calculated as in Equation (4). [37].
(4)$$\mathrm{MR}=\frac{m_{t}-m_{e}}{m_{o}-m_{e}}=\frac{8}{\pi^{2}}\sum_{n=1}^{\infty }\frac{1}{{\left(2n-1\right)}^{2}}\exp\left[-\frac{{\left(2n-1\right)}^{2}\pi^{2}D_{eff}t}{4L^{2}}\right]$$

For long drying durations, only the first term on the right side of the equation is taken into consideration, and the equation takes the form of Equation (5).
(5)$$\mathrm{MR}=\frac{m_{t}-m_{e}}{m_{o}-m_{e}}=\;\frac{8}{\pi^{2}}\;\exp\left[-\frac{\pi^{2}D_{eff}t}{4L^{2}}\right]$$

Here, *D_eff_* represents effective diffusivity (m^2^ s^−1^) and *L* represents the half-thickness (m) of the sample.

When the effects of microwave power on the effective diffusion coefficient are examined, a line is obtained and the value of the activation energy is calculated from the slope of this line. Activation energy is defined by the water molecules passing through the energy barrier when there is moisture transfer within the product. The small values of the activation energy give higher moisture diffusion values in the drying process. Reduction in the activation energy of a process is caused by the increase in the water molecules average energies [37]. The activation energy was estimated using the Arrhenius equation (Equation (6)). In order to determine the *D_eff_* value, changes of the *ln* (MR) values depending on the time were drawn as a graph, and the *D_eff_* values of the slopes of the lines that were obtained were calculated with Equation (7) [38].
(6)$$D_{eff}=D_{o}\exp\left(-\frac{E_{A}m}{P}\right)$$

From Equation (5), a plot of ln MR versus time gives a straight line with a slope of:(7)$$\mathrm{Slope}=-\frac{\pi^{2}D_{eff}}{4L^{2}}$$

*D_o_* is a constant equivalent to diffusivity at the infinite temperature (m^2^ s^−1^), *E_A_* is activation energy (W g^−1^), and *m* (g) is the mass of the raw product.

#### 2.3.5. Specific Energy Consumption and Drying Energy Performance

The energy consumption of the microwave dryer is calculated from Equation (8), and the specific energy consumption is calculated from Equation (9) [39].
(8)$$E_{c}=P\times t$$
(9)$$E_{s}=\frac{P\times t\times 10^{-6}}{m_{ev}}$$

*E_s_* is the specific energy consumption (MJ kg water^−1^), *P* is the microwave power (W), *t* is the drying time (s), and *m_ev_* is the total mass of evaporated water (kg).

#### 2.3.6. Color Change in the Product, Scanning Electron Microscopy (SEM) and Product Temperature Measurements

Colorimetric parameters were used as shown in Equations (10) and (11) in order to characterize the color changes in the product. The measurements were based on the L*a* b* color scale developed by CIE in 1976. The L* value, which changes from black to white, gives the gloss value and can take values ranging from 0 to 100 depending on the measured color [17]. The parameter a* takes positive and negative values (reddish and greenish), whereas the parameter b * takes positive and negative values (yellowish and bluish) [27].
(10)$$\Delta E=\sqrt{{\left(\Delta L\right)}^{2}+{\left(\Delta a\right)}^{2}+{\left(\Delta b\right)}^{2}}$$
(11)$$\Delta L=L^{*}-{\;L}_{o}\;\Delta a=a^{*}-{\;a}_{o}\;\Delta b=b^{*}-{\;b}_{o}$$
where ΔE is total color change.

In order to have a grasp of the changes occurred in the internal structures of the persimmon samples during the drying process, the observations were made on an electron microscope (SEM). FEI brand (QUANTA FEG 250) scanning electron microscope was used to view the microstructure of the dried products. It was sliced by a razor at dimensions of 1 cm^2^ without damaging the upper side of the sample and was photographed.

The change in the product temperature during the drying process was measured by the usage of the FLIR Ex E6 brand (Estonia) thermal camera. Firstly, thermal images of the product were taken on its wet state at the room temperature in order to compare the temperature of the product. Afterwards, thermal imaging was taken during drying within 30 s for each of its slices at certain periods.

#### 2.3.7. Statistical Analyses

The quantitative data were expressed as mean values. The results were analyzed using a factorial design with analysis of variance (ANOVA). The Tukey’s test was applied to determine if the differences were significant. All statistical analyses were performed with SPSS (PASW Statistics 18, SPSS In., Chicago, IL, USA). Differences with *p*-values less than 0.05 were considered significant.

## 3. Results and Discussions

### 3.1. Drying Characteristics

The drying characteristics of the sliced persimmon samples of different sizes are given in Figure 2, Figure 3 and Figure 4. As can be observed, the drying rate is affected by microwave drying power depending on the increase in the drying power. Moisture loss is accelerated and drying time is shortened. Depending on the drying powers, the drying process took 28.5, 6.23, 5.22, 3.57 min at 5 mm slice thickness, 33, 8.5, 4, 4.2 min at 7 mm slice thickness, 27, 8.1, 6.6, 5.43 min, at 9 mm slice thickness, 27, 8.1, 6.06 and 5.43 min, respectively. Except for the drying durations at 120 W, it is observed that drying durations are close to each other. When taking a look at the literature, Table 1 is observed. Therefore, the present study has shown itself to be advantageous in terms of time.

### 3.2. Determination of Effective Diffusivity and Activation Energy

Table 2 shows the increase in diffusion coefficients by increasing the thicknesses of the persimmon slices. It also increases as the microwave power value increases. The increase in microwave power also causing an increase in effective diffusivity constant is stated in Zarein et. [38]. Effective diffusivity constant was calculated from 2.97 × 10^−8^ m^2^ s^−1^ at 120 W to 4.63 × 10^−6^ m^2^ s^−1^ at 600 W. The values calculated in this study were found to be higher than the effective diffusion coefficient values that were calculated by Doymaz [5] who dried persimmons by heated air. As shown in Figure 3, the activation energy was calculated from the change of *LnD_eff_* according to *m*/*P*. *LnD_eff_* is the logarithmic expression of Equation (6). Activation energy according to the slice thickness was estimated at 32.82, 18.64 and 12.80 W g^−1^ respectively. Apparently, the activation energy decreased with increasing slice thickness.

As seen in Figure 4, drying duration and drying rate decreased with the moisture content. At the first period of the drying, as the moisture value is high, the absorbed power is also high. As power being high will affect polar molecules in the product more and produce a higher heat [40]. Therefore, the drying process, in the beginning, was fast, and as the moisture decreased, the drying took place slower in the later periods. This shows us that diffusion is a mechanism that regulates moisture movement [21]. The drying rate increased with an increase in microwave power. Doymaz [5], Chahbani et al. [41] and Hanif et al. [6] also stated that the drying rate increases with an increase in the microwave power in their own studies. The average drying rates of dried persimmons at varying powers were found to be; 0.10, 0.43, 0.50 and 0.78 kg water kg dry matter^−1^ min^−1^ at 5 mm slice thickness, 0.11, 0.36, 0.70 and 0.71 kg water kg dry matter^−1^ min^−1^ at 7 mm thickness, 0.11, 0.37, 0.58 and 0.50 kg water kg dry matter^−1^ min^−1^ at 9 mm thickness, respectively. Experimental conditions and products not being homogeneous cause changes in the rate of the drying.

Energy consumption values were recorded by an energy counter. Energy consumption and specific energy consumption values were shown in Figure 5a,b. At different powers, it was measured that at 5 mm slice thickness, 0.065–0.085 kWh, at 7 mm slice thickness, 0.05–0.1 kWh, and at 9 mm slice thickness, 0.07–0.09 kWh, respectively. As the applied microwave power increased, the drying duration and energy consumption got reduced [42]. This decrease in the microwave energy results in less power production and longer drying durations. Thus the time required for the water in the material to reach the evaporation temperature increases and the energy spent on evaporation decreases. Specific energy consumption values for the selected forces were measured between 14.80–20.34 MJ kg^−1^ at 5 mm slice thickness, 14.79–27.66 MJ kg^−1^ at 7 mm slice thickness, and 12.18–19.89 MJ kg^−1^ at 9 mm slice thickness, respectively. Shrinkage % values of the persimmons that were dried at different powers according to their thickness are given at Figure 5c. While shrinkage increases at 5 mm slices with drying power value, it gets more complicated at different slice thicknesses. This is due to the fact that the water content of the material is not homogeneous. The minimum shrinkage was observed at 460 W at 7 mm, and the maximum shrinkage was observed at 600 W at 5 and 7 mm. 

According to variance analyses, results showed that effects of slice thickness on specific energy consumption (*E_s_*) values for samples were found to be significant (*p* < 0.05). Samples that were sliced into 7 mm, and another two samples that were 5 and 9 mm thickness, were found in different groups in respect of specific energy consumption. Effects of slice thickness on energy consumption values for all samples were found to be insignificant (*p* > 0.05). Effects of slice thickness on shrinkage values between samples that were sliced into 5 mm thickness, while another two samples that were sliced into 7 and 9 mm were found significant (*p* < 0.05).

When microwave power levels were considered, effects of power levels on energy consumption, specific energy consumption, percentage shrinking values were found significant (*p* < 0.05). Samples that were dried using 120 W and other three power levels were found in different groups in respect of energy consumption and specific energy consumption (*p* < 0.05), while samples that were dried using 600 W and other three power levels were found in different groups in respect of shrinkage (*p* < 0.05).

### 3.3. Color and SEM Analysis

The gloss and total color changes of the product that is dried in the microwave drying process are shown in Table 3. Although a comparison could not be made on the effects of the power change on color change, it has been observed that the color changes in the products lower as the slice thicknesses increase. The color and the volumetric changes of the persimmons that were cut into different sizes can be seen in Figure 6 through the photographs. The least gloss and total color change were observed at 600 W at 9 mm. When examining these changes, the waiting period of the product after it was harvested was extremely important. When the product is kept for too long, the homogeneous structure of the product deteriorates and the product becomes softer. This significantly changes the drying kinetics of the product. While the product maintains its shape integrity when the drying was done shortly after the harvest, the product could undergo deteriorations in its shape and structure if it is kept for too long. Burns can occur in some regions of the dried product. Due to the fact that the liquid content of this product is not homogeneous and there are pores due to the core in its center, burns occur in these regions until the product decreases to the desired moisture level according to the power value applied to the product. At lower power levels, when drying duration was long, it increased the burned regions.

According to variance analyses, effects of slice thickness on ΔL values for all samples were found to be insignificant (*p* > 0.05), while effects of slice thickness on ΔE values between samples that were sliced into 9 mm thickness and samples were found to be significant (*p* < 0.05). When microwave power levels were considered, effects of power levels on ΔL values between samples dried at 600 W and other power levels were found to be significant (*p* < 0.05), while effects of power level on ΔE values for all samples were found to be insignificant (*p* > 0.05).

By taking a 2 mm long, 1 mm wide piece between the dried persimmons center and corner, the microstructure of the sample was analyzed in an SEM device. The microstructure of the sample was determined by magnifying 1000× and is shown in Figure 7, Figure 8 and Figure 9 for each drying thickness. While the pores are shown in black solid phases are shown in white. As the power level increased, the number of the pores also increased and the pores got larger for 5 mm thickness. Similar observations were made and stated by Dong et al. [43] in their study regarding green coffee beans. Increasing power level caused increasing temperature and vapor pressure in the structure. Large pores were formed due to the steam outlet. However, it is not the case for 7 mm and 9 mm thicknesses. As microwave power increased, there was an increase in pores, but as the slice thickness increased, the pores shrunk. This was due to the penetration of energy into the product. The fact that microwave energy did not penetrate into the product decreased evaporation and caused the size of pores to shrink. It was also related to the homogeneity of the product. As shown in Figure 7, Figure 8 and Figure 9, the material loses its homogeneity.

### 3.4. Thermal Images and Temperature Measurements

A scale showing the minimum and maximum temperature values of samples is displayed in Figure 10, Figure 11, Figure 12, Figure 13, Figure 14, Figure 15, Figure 16, Figure 17, Figure 18, Figure 19, Figure 20 and Figure 21. High energy absorption and high drying rates lead to overheating in certain regions during drying. Overheating results in regional burns and makes it difficult to control [44]. In Figure 6, these regional burns can be seen in the final product. As the microwave power increased, the temperature of the final product increased. In certain regions, the color seems to be yellow. These regions are the areas where the product had tearing on its edges and where its circularity was disrupted.

## 4. Conclusions

In this study, the most suitable drying power for 5 mm slice thickness was determined as 600 W in terms of time. Drying operation for 600 W power lasted 3.57 min. In terms of energy consumption, minimum value of 0.05 kWh for 7 mm thickness was measured at 460 W. The drying time and energy consumption decreased considerably with an increase in the microwave power level. This can be explained with increasing of drying time with decreasing of microwave power level, and this caused an increase in consumed energy amount. 

Specific energy consumption values were measured for 120 W, 350 W, and 460 W microwave power level as between 14.80–20.34 MJ kg^−1^ for 5 mm, 14.79–27.66 MJ kg^−1^ for 7 mm, and 12.18–19.89 MJ kg^−1^ for 9 mm slice thicknesses, respectively.

The minimum shrinkage was observed at 460 W at 7 mm, and the maximum shrinkage was observed at 600 W at 5 and 7 mm.

Effective diffusion coefficient was changed from 2.97 × 10^−8^ m^2^ s^−1^ at 120 W to 4.63 × 10^−6^ m^2^ s^−1^ at 600 W. The activation energy values according to the slice thicknesses namely 5, 7, and 9 mm were estimated as 32.82, 18.64, and 12.80 W g^−1^, respectively. As the microwave power increased, the diffusion coefficient also increased.

Drying rate increased as the microwave power increased. At the beginning of the process, the drying occurred fast, while in later periods it occurred slowly as the moisture rate decreased.

Color change, which is one of the criteria in terms of quality in dried food products, with the values of 13.25 (ΔL) and 24.32 (ΔE), were observed at the minimum slice thickness of 9 mm at 600 W. While lower microwave power can cause a lower drying temperature and lower drying rate, higher microwave power can lead to an undesirable higher temperature, increasing the unstable distribution of microwave energy and damaging the color quality of the product. If it is supported by conventional drying, a more acceptable appearance can be obtained in terms of color properties.

When taking a look at the microstructure, as the power level increased, the number of pores also increased, and growths occurred in the pores.

At the end of the experiments, the temperature of the product was measured at 87.6 °C minimum and as 155 °C at maximum. The minimum heating was found to be 87.6 °C at 120 W power for 5 mm slice thickness. Thermal analyses have shown that the effect of slice thickness and power level on the product are important.

## Figures and Tables

**Figure 1 foods-08-00084-f001:**
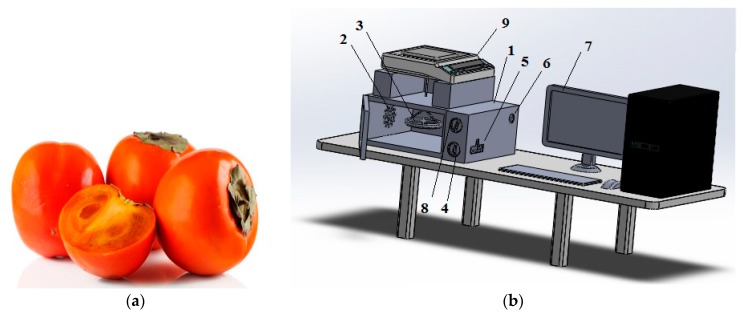
(**a**) Trabzon persimmon, (**b**) Microwave Drying System (1: Microwave oven, 2: Ventilation holes, 3: Tray, 4: Timer, 5: Magnetron, 6: Fan; 7: Computer, 8: Power switch, 9: Scales).

**Figure 2 foods-08-00084-f002:**
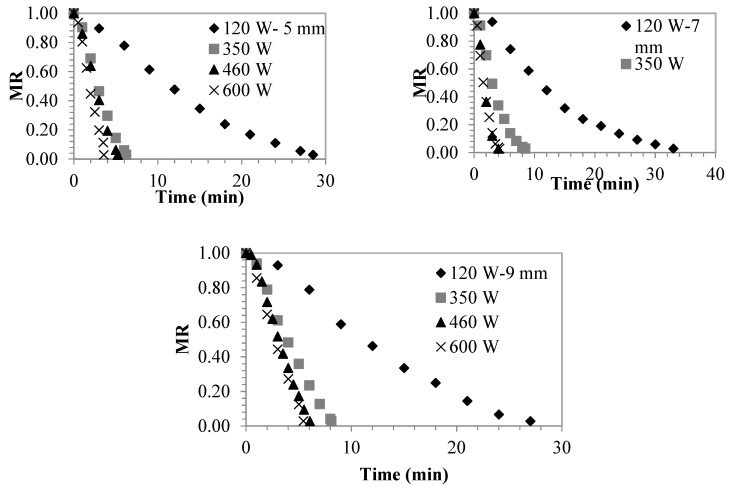
Changes in the moisture content based on time of the persimmons dried in varying microwave powers and varying slice thicknesses.

**Figure 3 foods-08-00084-f003:**
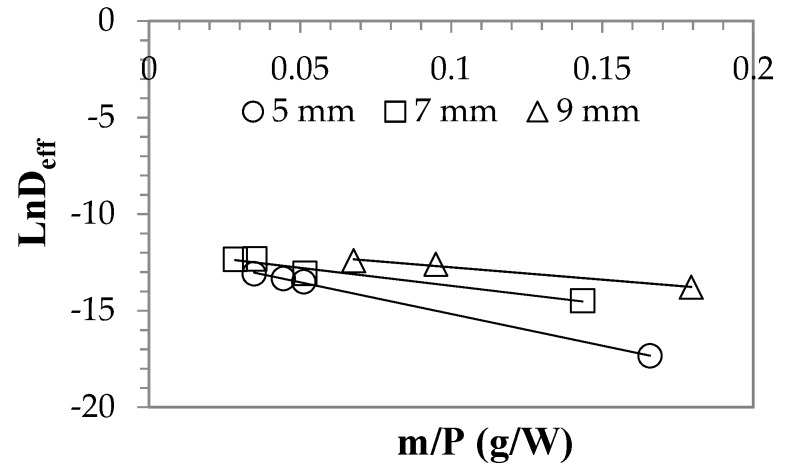
Microwave powers effect on effective diffusion coefficient.

**Figure 4 foods-08-00084-f004:**
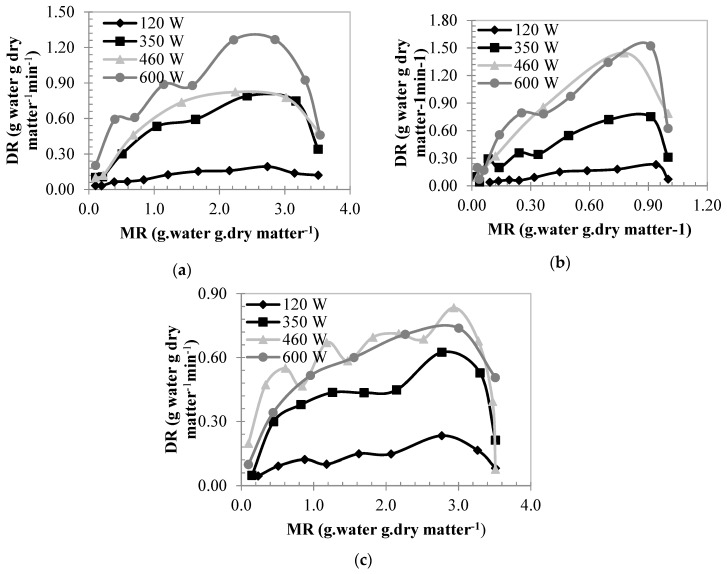
Variation of the drying durations with the changes in the drying rate at different powers. (**a**) 5 mm, (**b**) 7 mm, (**c**) 9 mm.

**Figure 5 foods-08-00084-f005:**
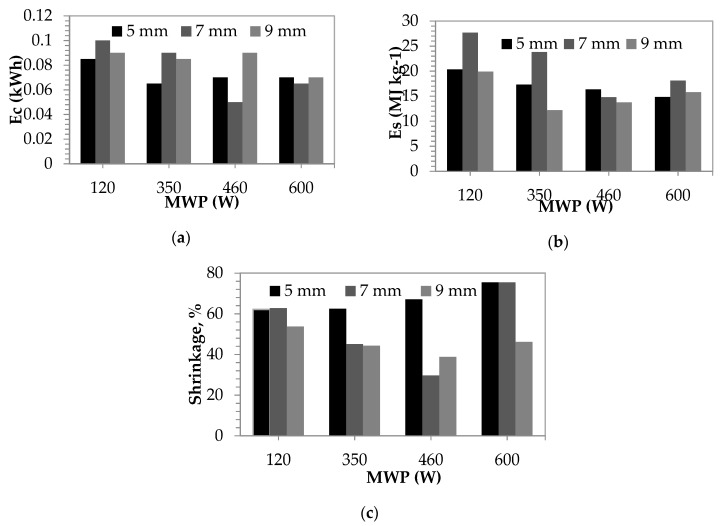
(**a**) Energy consumption values of the microwave dryer, (**b**) the specific energy consumption of the persimmon at varying powers and slices during the drying, (**c**) percentage shrinking of the persimmon during the drying.

**Figure 6 foods-08-00084-f006:**
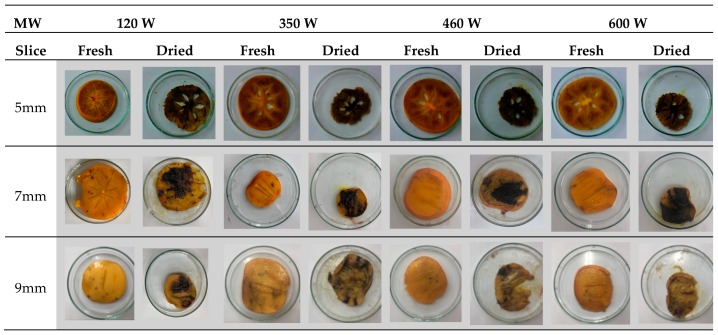
Fresh and dried pictures of persimmon.

**Figure 7 foods-08-00084-f007:**
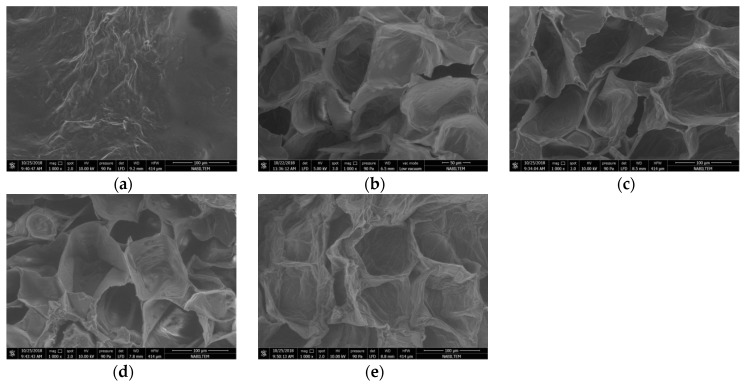
SEM (Scanning Electron Microscope) images of persimmon with 5 mm: (**a**) fresh persimmon; (**b**) dried persimmon at 120 W; (**c**) dried persimmon at 350 W; (**d**) dried persimmon at 460 W; (**e**) dried persimmon at 600 W.

**Figure 8 foods-08-00084-f008:**
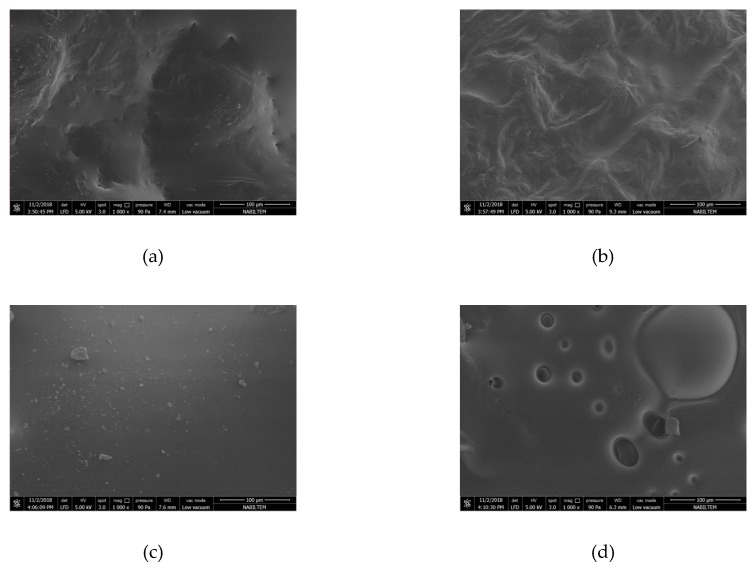
SEM (Scanning Electron Microscope) images of persimmon with 7 mm: (**a**) dried persimmon at 120 W; (**b**) dried persimmon at 350 W; (**c**) dried persimmon at 460 W; (**d**) dried persimmon at 600 W.

**Figure 9 foods-08-00084-f009:**
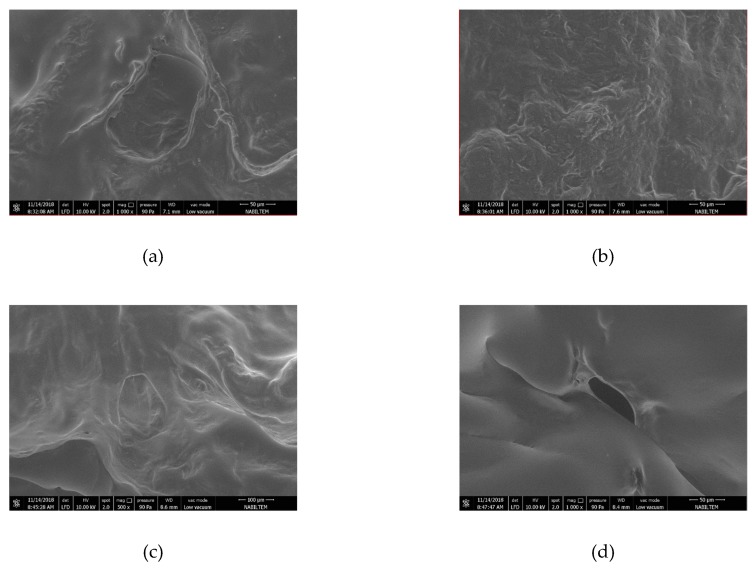
SEM (Scanning Electron Microscope) images of persimmon with 9 mm: (**a**) dried persimmon at 120 W; (**b**) dried persimmon at 350 W; (**c**) dried persimmon at 460 W; (**d**) dried persimmon at 600 W.

**Figure 10 foods-08-00084-f010:**
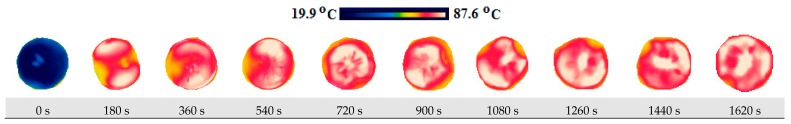
Thermal images of persimmon with 5 mm at 120 W.

**Figure 11 foods-08-00084-f011:**
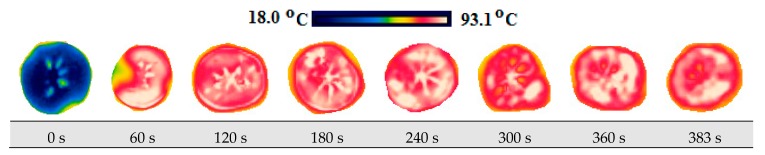
Thermal images of persimmon with 5 mm at 350 W.

**Figure 12 foods-08-00084-f012:**
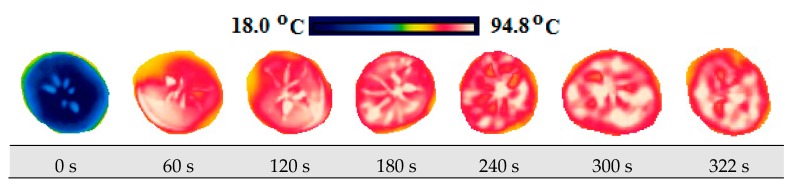
Thermal images of persimmon with 5 mm at 460 W.

**Figure 13 foods-08-00084-f013:**
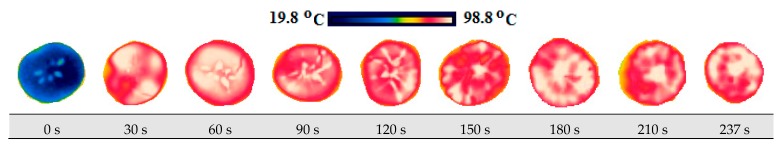
Thermal images of persimmon with 5 mm at 600 W.

**Figure 14 foods-08-00084-f014:**
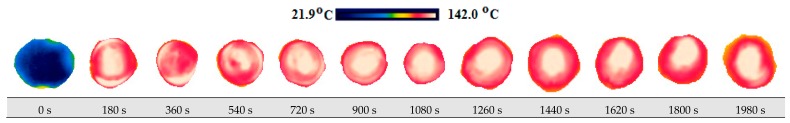
Thermal images of persimmon with 7 mm at 120 W.

**Figure 15 foods-08-00084-f015:**
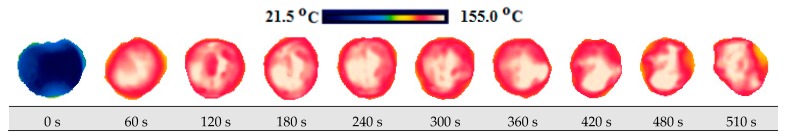
Thermal images of persimmon with 7 mm at 350 W.

**Figure 16 foods-08-00084-f016:**
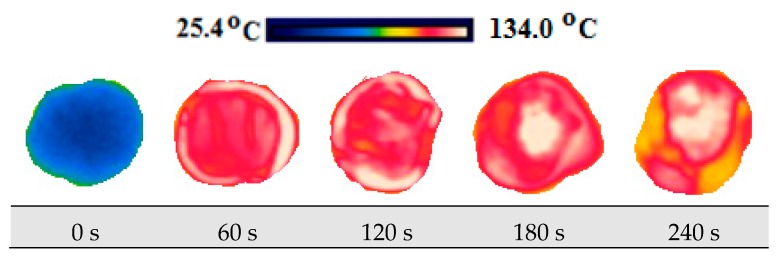
Thermal images of persimmon with 7 mm at 460 W.

**Figure 17 foods-08-00084-f017:**
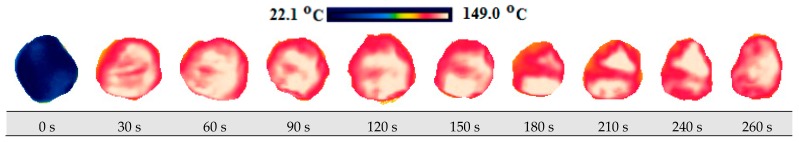
Thermal images of persimmon with 7 mm at 600 W.

**Figure 18 foods-08-00084-f018:**
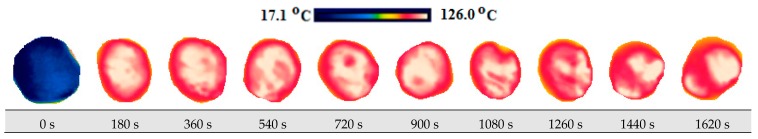
Thermal images of persimmon with 9 mm at 120 W.

**Figure 19 foods-08-00084-f019:**
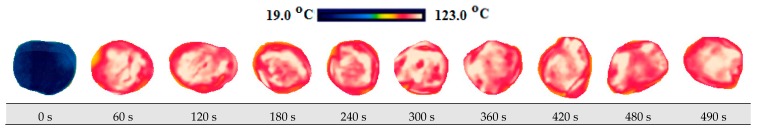
Thermal images of persimmon with 9 mm at 350 W.

**Figure 20 foods-08-00084-f020:**
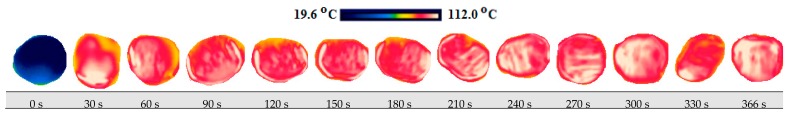
Thermal images of persimmon with 9 mm at 460 W.

**Figure 21 foods-08-00084-f021:**
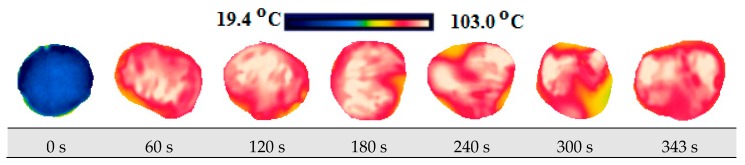
Thermal images.

**Table 1 foods-08-00084-t001:** Comparison of microwave drying with other drying methods.

Drying Type	Drying Time	References
microwave drying	3.57–28.5 min	this study
solar drying	16–22 h	[6]
convective drying	15–40 h	[4]

**Table 2 foods-08-00084-t002:** The diffusion coefficients and activation energy values of the persimmon slices at different powers.

MWP (W)	*D_eff_* (m^2^ s^−1^)		
	5 mm	7 mm	9 mm
120	2.97 × 10^−8^	5.15 × 10^−7^	1.03 × 10^−6^
350	1.37 × 10^−6^	2.12 × 10^−6^	3.39 × 10^−6^
460	1.63 × 10^−6^	4.36 × 10^−6^	4.10 × 10^−6^
600	2.07 × 10^−6^	4.47 × 10^−6^	4.63 × 10^−6^
*E_a_* (W g^−1^)	32.82	18.64	12.80

MWP is microwave power (W).

**Table 3 foods-08-00084-t003:** Color parameter results.

Slice	Parameters	120 W	350 W	460 W	600 W
5 mm	ΔL	33.53	23.83	14.91	16.72
	ΔE	46.09	34.19	20.28	29.64
7 mm	ΔL	28.49	22.19	28.30	16.36
	ΔE	47.63	29.91	49.26	25.14
9 mm	ΔL	17.25	21.56	21.33	13.25
	ΔE	17.53	28.84	29.34	24.32

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
