# Peer review of "Effect of Microwave Drying on the Drying Characteristics, Color, Microstructure, and Thermal Properties of Trabzon Persimmon"

_foods, 2019, doi:10.3390/foods8020084_

Round 1
Reviewer 1 Report
Introduction
In the first part of introduction please explain why the author choose persimmon for drying and why is it necessary to optimize persimmon drying.
Furthermore, in the introduction the author has to focus on microwave drying of persimmon. The state of art about mechanism is too long.
Infact the author wrote just two sentence about persimmon drying in lines 67-70. Please improve.
lines 33-36: please insert some other new references about the advantages and disadvantages of drying, for example: Adiletta, G., Russo, P., Proietti, N.,Capitani, D.,Mannina, L.,Crescitelli, A., Di Matteo, M.Characterization of pears during drying by conventional technique and portable non invasive NMR. 2015 Chemical Engineering TransactionsVolume 44, May 2015, Pages 151-156.
Material and methods
Please insert a picture of the selected persimmon cultivar.
line 83: "and 9 mm (5-7.2 cm diameters.)" of thickness.
line 86: "cavity dimensions of 419x428x245", please insert units.
Please check the equation 1a. It seems wrong if the author considered wet metter
Line 171: change calorimetric parameter to colorimetric parameters.
Line 177: the author has to explain all CIELAB parameters not only L*.
Line 184: please explain the treatment of samples before SEM analysis.
Results and discussion
lines 213-214: how can the author explain the results of activation energy?
SEM analysis
The SEM pictures for 5 mm of thickenss are totally different from 7 and 9 mm. In the latter the pores are absent. Why? The author said that as the power level increased, the number of
pores also increased. It is not cottect. Please explain.
Author Response
Response to Reviewer 1 Comments
Dear Reviewer,
First of all, I want to thank you for your kind suggestions and comments for my article. The publication of the article is very important for my academic advancement. I took your suggestions into consideration and did the necessary corrections in the article. I hope that all of them are enough for you. The corrections are given below.
Best regards.
Point 1: In the first part of introduction please explain why the author choose persimmon for drying and why is it necessary to optimize persimmon drying.
Response 1: I have explained in the last part of introduction.
Point 2: Furthermore, in the introduction the author has to focus on microwave drying of persimmon. The state of art about mechanism is too long.
Response 2: Thank you for your suggestion. In general I explained brief information.
Point 3: Infact the author wrote just two sentence about persimmon drying in lines 67-70. Please improve.
Response 3: Sentences number has been increased. New references have been added.
Point 4: lines 33-36: please insert some other new references about the advantages and disadvantages of drying, for example: Adiletta, G., Russo, P., Proietti, N.,Capitani, D.,Mannina, L.,Crescitelli, A., Di Matteo, M.Characterization of pears during drying by conventional technique and portable non invasive NMR. 2015 Chemical Engineering TransactionsVolume 44, May 2015, Pages 151-156.
Response 4: Recommended reference was added.
Point 5: Please insert a picture of the selected persimmon cultivar.
Response 5: It has been added as Figure 1a.
Point 6: line 83: "and 9 mm (5-7.2 cm diameters.)" of thickness.
Response 6: It has been correct.
Point 7: line 86: "cavity dimensions of 419x428x245", please insert units.
Response 7: It has been correct.
Point 8: Please check the equation 1a. It seems wrong if the author considered wet metter.
Response 8: The suggestion was determined and it was concluded that I decided that it was correct.
Point 9: Line 171: change calorimetric parameter to colorimetric parameters.
Response 9: It has been correct.
Point 10: Line 177: the author has to explain all CIELAB parameters not only L*.
Response 10: This was explained in the section of 2.3.6. “The parameter a* takes positive and negative values (reddish and greenish), whereas the parameter b * takes positive and negative values (yellowish and bluish)”.
Point 11: Line 184: please explain the treatment of samples before SEM analysis.
Response 11: This was explained in the section of 2.3.6. “It was sliced by razor at dimensions of 1 cm2 without damaging the upper side of the sample and was photographed.”
Point 12: lines 213-214: how can the author explain the results of activation energy?
Response 12: This was explained in the section of 3.2. “Apparently, the activation energy decreased with increasing slice thickness.”
Point 13: The SEM pictures for 5 mm of thickenss are totally different from 7 and 9 mm. In the latter the pores are absent. Why? The author said that as the power level increased, the number of pores also increased. It is not correct. Please explain.
Response 13: This was explained in the section of 3.3. “Increasing power level caused increasing temperature and vapor pressure in the structure. Large pores were formed due to the steam outlet. The same situation was not valid for 7 mm and 9 mm thicknesses. As microwave power increased, there was an increase in pores but as the slice thickness increased the pores shrunk. This was due to the penetration of energy into the product. The fact that microwave energy did not penetrate into the product decreased evaporation and caused the size of pores to shrink. It was also related to the homogeneity of the product.”
Reviewer 2 Report
The article “Effect of Microwave Drying Characteristics, Color, Microstructure, and Thermal of Trabzon Persimmon“ deals with observations of various product and process properties during the drying of persimmon slices with microwaves. The article gives almost no new insights in microwave drying. Most of the findings are trivial or extremely specific for this product and this special experimental setup. Therefore, this article is only of minor interest for the scientific community. Nevertheless, if it would be written in a clear and comprehensive manner one could publish it to show these specific results. But unfortunately there are many poor expressions and incomplete descriptions.
The main points are:
- There is no statistical information on the experiments.
- The author claims that it is very important to take into account the time from harvest to experiment. But he does not give this information.
- The results of the drying experiments are described in detail, while a discussion of the results is almost non-existent. To be of scientific value, proposed reasons of the observed phenomena should be added and appropriate (new) conclusions should be drawn.
- Trivial results can be omitted or should be clearly stated like: “ As expected with increasing microwave power level the drying time decreases…”. These include e.g. the findings that an increase in power results in shorter drying times or that microwave drying is faster than solar drying.
- Figures 6-21 contain huge amounts of data. These results are not appropriately described in the text.
- Assumptions made in Chapter 2.3.4 are incomplete (see chapter 4 Handbook of Industrial Drying) and in part not correct. Add justifications for false assumptions.
- The finding on structure are not based on comprehensible results
- The findings on temperature (fig 10 to 21) are not comprehensible at all. From the colors no temperatures can be derived from the reader. This part is completely senseless.
- The conclusions are a list of specific findings or trivial statements. There is no critical discussion of the findings and no new scientific finding or even hypothesis.
General comments
- Check grammar, wording, expressions, especially punctuation in the article by a professional!
- Explanations for all variables need to be added.
- The units for the different variables should be formatted according to ISO Standards.
- Eliminate citations of secondary literature.
- Figures should be uniform in their depiction. If an interpolation between the measured points is used (which is not always appropriate), the corresponding equation should be added. The axes are not labeled according to ISO Standards.
- Formatting of equations can be improved. Dots are no multiplication signs. Cite original sources of equations.
- Organize big chunks of data in tables, e.g. L196
Specific comments
- L13: At this point it is not clear what the mm information is
- L18: Add specific findings about structure and heating in the introduction
- L36-38 and Line 40: You cannot make this general statement based on this single Literature.
- L43: I am sure that there is much more Literature. Cant you cite a review or a book chapter...?
- L55: This is not true. You have always a constant rate period first!
- L85: Fig. 1 is only a sketch.
- L91...: What do you mean with intervals? Weighing intervalls?
- L93: 2450 Hz should probably be 2450 MHz
- L94: there are no changes before you start the experiment
- L100: What exactly have you measured?
- L103: Figure contains irrelevant desk and computer
- L114: Equilibrium moisture content is not shown in the formula for MR
- L127: How did you calculate this? What have you measured?
- L131-132: This is not a sentence.
- L147: Didn’t you use the specific microwave power?
- L171: Sorry but this is just a hypothesis. And trust me: it is not true! You are burning your product. There are definitely more changes than only carotene oxidation!
- L180: Definition of symbols is missing
- L181: How was the sample preparation?
- L186: Who is we? I see only one author.
- L194: Give a table with the data. On how many slices this data is based on? How often have you repeated the experiment?
- L199 & L200: Time advantage is trivial
- L224: Why does it show this?
- L232: Figure 4: Interpolation between points is not valid
- L249: Not homogenous material: influence of the non-homogenous structure needs to be quantified in order to compare drying processes with different process parameters
- L265: 640 W ?
- L266: This duration between harvest and experiment is completely neglected in the other parts of the article.
- L270: It is typically burned because the microwaves are focussed and heat the product differently at different positions.
- L293: The single pictures are no evidence of larger pores. Pores seem to be of constant diameter (Fig 7). I cannot see any analyzable pores with statistical relevance in Fig 8 and 9.

Author Response
Response to Reviewer 2 Comments
Dear Reviewer,
First of all, I want to thank you for your kind suggestions and comments for my article. The publication of the article is very important for my academic advancement. I took your suggestions into consideration and did the necessary corrections in the article. I hope that all of them are enough for you. The corrections are given below.
Best regards.
Point 1: The author claims that it is very important to take into account the time from harvest to experiment. But he does not give this information.
Response 1: In this process, the persimmon is to lose its hardness and homogeneity.
Point 2: The results of the drying experiments are described in detail, while a discussion of the results is almost non-existent. To be of scientific value, proposed reasons of the observed phenomena should be added and appropriate (new) conclusions should be drawn.
Response 2: In each section, the studies in the literature were compared. Less research was done on persimmon drying in literature.
Point 3: Trivial results can be omitted or should be clearly stated like: “ As expected with increasing microwave power level the drying time decreases…”. These include e.g. the findings that an increase in power results in shorter drying times or that microwave drying is faster than solar drying.
Response 3: Thanks for your suggestion.
Point 4: Figures 6-21 contain huge amounts of data. These results are not appropriately described in the text.
Response 4: Figures 6-21 are briefly explained in related to sections.
Point 5: Assumptions made in Chapter 2.3.4 are incomplete (see chapter 4 Handbook of Industrial Drying) and in part not correct. Add justifications for false assumptions.
Response 5: Thanks for your suggestion.
Point 6: The findings on temperature (fig 10 to 21) are not comprehensible at all. From the colors no temperatures can be derived from the reader. This part is completely senseless.
Response 6: The colors in figure 10-21 are clearly visible. It is compatible with scale.
Point 7: L13: At this point it is not clear what the mm information is.
Response 7: It has been added as “mm”.
Point 8: L18: Add specific findings about structure and heating in the introduction.
Response 8: There are specific findings about structure and heating in the introduction.
Point 9: L36-38 and Line 40: You cannot make this general statement based on this single Literature.
Response 9: They are added as references 5, 10, 11.
Point 10: L43: I am sure that there is much more Literature. Cant you cite a review or a book chapter...?
Response 10: Thanks for your suggestion.
Point 11: L55: This is not true. You have always a constant rate period first!
Response 11: There is always a constant rate period first. I mentioned that the fastest drying was in the second stage.
Point 12: L85: Fig. 1 is only a sketch.
Response 12: I've added this Figure to add a bit of visuality.
Point 13: L91...: What do you mean with intervals? Weighing intervals?
Response 13: I mean the period in which the product weight was recorded during drying.
Point 14: L93: 2450 Hz should probably be 2450 MHz
Response 14: It has been corrected as 2450 MHz
Point 15: L100: What exactly have you measured?
Response 15: The energy consumed at the end of drying was determined by the energy meter.
Point 16: L103: Figure contains irrelevant desk and computer
Response 16: The computer was used to transfer data throughout the experiment.
Point 17: L114: Equilibrium moisture content is not shown in the formula for MR
Response 17: Equilibrium moisture content (Eq. 1b) was added.
Point 18: L127: How did you calculate this? What have you measured?
Response 18: The diameter and thickness of the date slices were measured and volume was calculated.
Point 19: L171: Sorry but this is just a hypothesis. And trust me: it is not true! You are burning your product. There are definitely more changes than only carotene oxidation!
Response 19: Thanks for your suggestion. The sentence is deleted.
Point 20: L180: Definition of symbols is missing
Response 20: Definition of symbols is added.
Point 21: L181: How was the sample preparation?
Response 21: It was sliced by razor at dimensions of 1 cm2 without damaging the upper side of the sample and was photographed.
Point 22: L186: Who is we? I see only one author
Response 22: It is corrected.
Point 23: L194: How often have you repeated the experiment?
Response 23: Drying tests were repeated three times for each experimental condition in order to minimize the uncertainties in the results.
Point 24: L199 & L200: Time advantage is trivial
Response24: Drying time is important in the drying sector. I mentioned that my work is advantageous from other drying methods.
Point 25: L224: Why does it show this?
Response 25: Diffusion is related to moisture movement. Additional information was given. This sentence can be removed.
Point 26: L265: 640 W ?
Response 26: It has been corrected as 600 W.
Point 27: L266: This duration between harvest and experiment is completely neglected in the other parts of the article.
Response 27: Yes, it was neglected. Its freshness cannot be preserved because a product from the market is used.
Point 28: L270: It is typically burned because the microwaves are focussed and heat the product differently at different positions.
Response 28: I agree with your suggestion.
Point 29: L293: The single pictures are no evidence of larger pores. Pores seem to be of constant diameter (Fig 7). I cannot see any analyzable pores with statistical relevance in Fig 8 and 9.
Response 29: I commented on the results of my work. Thank you for your suggestion.
Round 2
Reviewer 1 Report
the author modified correctly the paper and put my suggestions.
Author Response
Dear Reviewer,
I want to thank you for your kind suggestions and comments for my article.
Best regards.
Reviewer 2 Report
Dear Autor,
thank you for the fast revision. You addressed mainly grammatical issues and some minor aspects. Unfortunately many of the main points of criticism are not addressed and with some of the changes new issues arise. Some examples:
The title is still inappropriate
„Effect of Microwave on … the Thermal Properties of Tr. Persimmon”
Thermal properties are e.g.: thermal conductivity, heat capacity. This is not content of the paper.
L38: The statement is still incorrect. It is not possible to reduce microorganisms to a level at which they cannot reproduce.
Comment: There is no statistical information on the experiments.
There is still no statistical information on the experiments.
Comment: L194: How often have you repeated the experiment
Only an answer for the reviewer. What are the errors? standard deviation…?
Comment: L127: How did you calculate this? What have you measured?
There is an explanation for the reviewer. But no explanation for the reader.
Comment: The author claims that it is very important to take into account the time from harvest to experiment. But he does not give this information.
No information for the reader added. The freshness will also influence your structure investigation and the drying time. How did you eliminate this influence?
Comment: The results of the drying experiments are described in detail, while a discussion of the results is almost non-existent. To be of scientific value, proposed reasons of the observed phenomena should be added and appropriate (new) conclusions should be drawn.
Some discussion is added to the structure analysis (SEM). But there is still no information on the pore size measurement and the statistics. Therefore, the conclusions that are drawn, cannot be understood.
Comment: L232: Figure 4: Interpolation between points is not valid
No change
Comment:Figures 6-21 contain huge amounts of data. These results are not appropriately described in the text.
There is no proper description added. No results deleted.
Comment: Assumptions made in Chapter 2.3.4 are incomplete (see chapter 4 Handbook of Industrial Drying) and in part not correct. Add justifications for false assumptions.
No change.
Comment: The findings on temperature (fig 10 to 21) are not comprehensible at all. From the colors no temperatures can be derived from the reader. This part is completely senseless.
Response 6: The colors in figure 10-21 are clearly visible. It is compatible with scale.
Answer: The scale is different for each figure. This is why the figures cannot be compared. And there is no proper scaling: It is impossible for the reader to find out which temperature is e.g. yellow or green…
This list is still not complete. This is not a proper revision of the manuscript.
Author Response
Dear Reviewer,
First of all, I want to thank you for your kind suggestions and comments for my article.
I took your suggestions into consideration and did the corrections in the article. The corrections are given below.
Best regards.
Response to Reviewer 2 Comments
Point 1: There is no statistical information on the experiments.
Response 1: Statistical analysis was done and the results were added to section 3.2 and 3.3.
Point 2: The finding on structure are not based on comprehensible results
Response 2: In the literature, there is no study in which the Trabzon persimmon is dried with microwave. The information and comparisons related to different products are given in the related sections.
Point 3: The conclusions are a list of specific findings or trivial statements. There is no critical discussion of the findings and no new scientific finding or even hypothesis.
Response 3: The results showed that microwave drying was advantageous in terms of time and energy consumption. It was emphasized how the microwave energy changed the microstructure of persimmon. It was also given with the thermal images how the temperature changes during the drying process.
Point 4: L94: there are no changes before you start the experiment
Response 4: It has been corrected.
Point 5: L131-132: This is not a sentence.
Response 5: It has been corrected.
Point 6: L147: Didn’t you use the specific microwave power?
Response 6: I did not use the specific microwave power.
Point 7: L232: Figure 4: Interpolation between points is not valid
Response 7: It is the graph drawn by the available data because of fact that drying time is short.
Point 8: L249: Not homogenous material: influence of the non-homogenous structure needs to be quantified in order to compare drying processes with different process parameters.
Response 8: Thanks for your suggestion. However, the subject you mentioned was not a matter of priority in this article. Therefore I did not examine this issue. You can be sure that I will also take this into consideration in future studies.
Point 9: L 13-14. This sentence requires editing.
Response 9: It has been corrected.
Point 10: L 17. Replace “drying speed” by “drying rate”. This concerns the whole manuscript.
Response 10: The article was rearranged according to your suggestion.
Point 11: L 25. Use italics for Latin names “(Diospyros kaki L)”
Response 11: It has been corrected.
Point 12: L 29. Persimmon cannot lower the cholesterol level unless it has been consumed. Please correct this sentence.
Response 12: It has been deleted.
Point 13: L 94 – 96. This sentence is not consistent. It is necessary to consistently mention the drying methods or the drying equipment.
Response 13: The test setup is laboratory scale and as seen in the Figure 1, it consists of dryer, precision scale, computer and measurement equipments. Technical information on these devices was given in section 2.2.
Point 14: L 94 – 96. “The reason why dried persimmon is used is that it is preferred widely and there are few studies regarding microwave energy”. This sentence should be divided into two sentences which need to be developed in order to sound meaningfully.
Response 14: The sentence was rearranged according to your suggestion.
Point 15: L 109. The time of storage in freeze conditions should be specified.
Response 15: The product was kept in the freezer for 1 day. This application was mentioned in Chapter 2.
Point 16: L 152. It is not clear whether the heterogeneity of microwave radiation was compensated if the product did not change its position?
Response 16: The product is fixed to the scale to detect weight losses. As the energy reaches the product directly by reflecting the product, there are regional differences. The aim of the article was to determine the drying parameters by keeping the product constant.
Point 17: L 196. Check whether the units are consistent as moisture content wet basis “m” was expressed in g water/g wet matter.
Response 17: I checked again and rearranged the formulas and units in the section 2.3.
Point 18: L 202. In which way the volume of the samples was measured? It was pycnometer or replacement method with using of toluene, flax seeds or other material?
Response 18: The diameter and thickness of the persimmon slices were measured and volume was calculated.
Point 19: L 391. The results shown in Figure 5 should contain statistical values to assess whether the mean values are representative. Anova results or at least standard deviations should be provided.
Response 19: All statistical analyses were performed with SPSS (PASW Statistics 18).
Point 20: L 426. Consider deleting “before drying”.
Response 20: Thanks for your suggestion. It is deleted.
Point 21: SEM images are only clear for samples with a thickness of 5 mm. For the remaining samples, representative images relating to one power will be sufficient. Only one image obtained for the fresh sample with a thickness of 5 mm is enough.
Response 21: I added fresh sample picture as you suggested. I want other images to remain in the text to be able to compare.
Point 22: In the section “Thermal images and temperature measurements” there are too many images which are not enough discussed. The discussion on the effect of microwave power and thickness of slices on the temperature course during drying would be more appropriate. This temperature course can be demonstrated using table or graph. On the other hand thermal images may concern only chosen stages of drying in order to present initial, final and maximal temperature of representative samples. In this way the number of figures could be limited.
Response 22: Microwave drying of persimmon has not been studied before. Since this study is original, it has been difficult to compare this product. A little comparison has been made. I believe that I made enough comparison in the temperature measurement and in our analyses outside SEM.
Point 23: The conclusions should be better presented in shorter and more comprehensive form. At present, conclusions are a kind of report without stressing the most important facts that bring some novelties relevant for drying technology and suggestions for practical applications. Try to improve this section as best as possible.
Response 23: The results obtained in the experiments and the proposed parameters are shown in conclusion section. These results emphasize that the drying system we use is advantageous on persimmon drying. Each analysis was explained shortly because of there are the different analyses.
Point 24: English writing style must be improved. The best option is to use the native speaker service. English native speaker with technical experience should correct the manuscript.
Response 24: English writing style rechecked and edited by English native speaker.